# Using Computer Vision to Track Facial Color Changes and Predict Heart Rate

**DOI:** 10.3390/jimaging8090245

**Published:** 2022-09-09

**Authors:** Salik Ram Khanal, Jaime Sampaio, Juliana Exel, Joao Barroso, Vitor Filipe

**Affiliations:** 1Research Center in Sports Sciences, Health Sciences and Human Development, CIDESD, Universidade de Trás-os-Montes e Alto Douro, 5000-801 Vila Real, Portugal; 2Institute for Systems and Computer Engineering, Technology and Science, INESC TEC, 4200-465 Porto, Portugal; 3Department of Sport Science, Biomechanics, Kinesiology and Computer Science, University of Vienna, 1150 Vienna, Austria

**Keywords:** computer vision, facial image analysis, color models, physical exercise intensity, heart rate

## Abstract

The current technological advances have pushed the quantification of exercise intensity to new era of physical exercise sciences. Monitoring physical exercise is essential in the process of planning, applying, and controlling loads for performance optimization and health. A lot of research studies applied various statistical approaches to estimate various physiological indices, to our knowledge, no studies found to investigate the relationship of facial color changes and increased exercise intensity. The aim of this study was to develop a non-contact method based on computer vision to determine the heart rate and, ultimately, the exercise intensity. The method was based on analyzing facial color changes during exercise by using RGB, HSV, YCbCr, Lab, and YUV color models. Nine university students participated in the study (mean age = 26.88 ± 6.01 years, mean weight = 72.56 ± 14.27 kg, mean height = 172.88 ± 12.04 cm, six males and three females, and all white Caucasian). The data analyses were carried out separately for each participant (personalized model) as well as all the participants at a time (universal model). The multiple auto regressions, and a multiple polynomial regression model were designed to predict maximum heart rate percentage (maxHR%) from each color models. The results were analyzed and evaluated using Root Mean Square Error (RMSE), F-values, and R-square. The multiple polynomial regression using all participants exhibits the best accuracy with RMSE of 6.75 (R-square = 0.78). Exercise prescription and monitoring can benefit from the use of these methods, for example, to optimize the process of online monitoring, without having the need to use any other instrumentation.

## 1. Introduction

The current technological advances have pushed the process of quantifying exercise intensity to higher levels in sports sciences. Monitoring physical exercise is essential in the process of planning, applying, and controlling loads for performance optimization and health [1,2,3]. The physiological responses vary depending upon the individual, the task, and the level of exertion during the physical exercise [4]. In steady-state continuous exercise, the level of physiological exertion is very low at the beginning and increases linearly with exercise intensity. In cases where intensity rises above the ventilatory and/or anaerobic threshold, the increase in the exertion is exponential. The changes of cardiovascular status during exercise and, broadly, the entirety of physiologic function, is reflected in heart rate (HR) responses [5,6], although body temperature, blood pressure, and blood lactate concentration can also quantify internal loads [7]. 

Exercise intensity can be estimated using objective and subjective measurements. The subjective measurements represent the level of exertion imposed by exercise through psychophysiological scales which require considerable individual familiarization and consume additional training time. Borg scale of perceived exertion [4] is one of the well-known examples of it. An important part of the research related to the measurement of the physical exercise intensity relies on the extraction of the physiological parameters such as heart rate [8,9], as the gold standard measure internal responses to the exercise. The previous studies demonstrated that the heart rate response to exercise is quite individual, whether it is moderate or vigorous exercise [10,11]. The HR response presents direct relation to the cardiac output during effort situation and can be obtained using a contact sensor technique or contact-less technique [12,13].

The use of wearables for monitoring biological signals during exercise is a trend that releases a novel perspective for an ecologic approach based on investigations in the sports sciences and medicine areas [14,15]. However, the contact sensors may present some drawbacks to users such as some discomfort during the exercise. Moreover, during movement, the devices can lack the contact with the skin, thus impairing data recordings. There is also a chance of failure in the data transmission from the sensor to the computing device, which might impair data sampling. The contactless sensors technology [16,17,18] and special methodologies for processing the contactless signals [19,20,21,22] are good alternatives to overcome the previous mentioned difficulties in estimating exercise intensity and energy expenditure preserving the ecological demands in applied science [12,23]. In this sense, facial features are the trending possibilities of the contactless estimation of physiological parameters [7,24,25]. Although facial expression of emotions is a cue that has been related to changes in physical and mental state [26,27,28,29,30], the use of facial color is an emerging technique in pattern recognition [31,32,33] that may provide a simple but innovative means of information that can be linked with exertion during exercise.

In the recent years, the exercise-related data are analyzed using several statistical and machine learning approaches which are proposed to classify and estimate the intensity of exercise and accumulated fatigue [34,35,36,37,38,39]. The outcome of research studies [34,35] suggested the strong correlation of facial color to heart rate. This evidence point out towards the potential of using the facial color analysis to estimate the physical exertion levels, however, there are no studies advancing towards the dynamics of gold standard measures as heart rate and perceived exertion, across the exercise. Despite a lot of research studies proposing various statistical approaches to estimate various physiological indices, to our knowledge, no studies have been found to investigate the relationship of facial color changes and increased exercise intensity.

Thus, in this paper, we aimed to describe the efficacy of various color models to facial color tracking in relating to heart rate dynamics during fatiguing exercise. Our investigation intended to find the potential statistical relationship between facial color changes and exercise intensity induced by heart rate. We believe that the deeper understanding of the relationship between the facial color changes and the exercise intensity is important not only the research perspective, but also practical application [40]. The autoregressive analysis is used as a statistical tool to investigate the relationship. The possibility to designed global regression equation is also investigated using a global regression model considering all subjects’ data at a time. The polynomial regression equation is also designed for global model. Standard error, t-statistics, p-values, multi-collinearity between each color component of all the color models are also taken into consideration. The outcome of this research contributes to design the better exercise model for the individual. Besides this, the automatic control of the exercise equipment based on the intensity of the exercise could be implemented. Moreover, the outcome of this research study can be useful for the accident control of the elderly people during physical exercise. 

The rest of this paper is organized as follows: Section 2 will review the works related to this study. Next, in Section 3, the proposed methods will be explained in detail. Then, in Section 4, the evaluation of the proposed approach through experimental results and discussion will be shown in Section 5. Finally, in Section 6, this paper will be concluded, and future work directions will be provided.

## 2. Related Works

One of the noticeable application fields of computer vision using facial expression and color analysis is drivers’ tiredness and fatigue detection. Many of them use color information from digital images to improve the algorithms [22,41,42]. This can be an important cue to link the fatigue and tiredness detection during the physical exercise. In face recognition related problems, the facial color information represented in the different color spaces is a valuable tool to improve the pattern detection [22,32,43,44,45,46,47]. Thus, it seems coherent that a facial expression detection might also be enhanced by considering the various color models [48,49]. The computer vision techniques showed to be feasible to detect perceived exertion [50,51]. However, there is a lack of information on the effectiveness of such non-contact techniques and its models, as the ones related to image processing, in describing the physiological process of fatigue during exercise when compared to gold standard contact sensors.

Digital images can be represented by the intensity of the color components present in each pixel through a range of various color channels. Recent research efforts have described the various color models rather than RGB (Red; Green; Blue) which may provide effective information for facial image processing such as face detection and recognition, etc. Various studies have been proposed to measure the physiological parameters including HR using alternative color spaces (HSV, YCbCr, Lab, etc.) instead of raw RGB. Leangwattana [52] proposed an approach to measure HR using each color com-ponent of RGB and HSV (Hue; Saturation; Value) and found that the H component and a third principal component of R provides better accuracy. Wang et. al. proposed a model to recognize the micro-expression and results showed that the performance of Tensor Independent Color Space (TICS), CIELab, and CIELub are better than those of RGB or gray [53,54]. Such efficacy of color models different than the RGB and gray for estimating levels of exertion along physical exercise is still unknown. The literature has shown that the facial color variation is related to the increase of exercise intensity, as redder skin due to tiredness from the physical exertion [45]. Dynamic appearance model of skin color built from in vivo and real time measurements of melanin and hemoglobin concentration has also been described to change according high-intensity exercise [43].

Several statistical and machine learning approaches were proposed recently to classify and estimate the intensity of exercise and accumulated fatigue. Timme [40] proposed a statistical approach using multilevel regression to analyze the facial action movement to predict exertion during incremental physical exercise. The authors concluded that there are significant changes in facial active with respect to exercise intensity. Not only the facial expression, facial color changes during exercise is also a potential cue to estimate the exercise intensity. Perrett [55] investigated the relationship between skin color, aerobic fitness (measured VO2 max), and body fat. The results suggested that there are significant changes in facial color with respect to the exercise intensity. Resting heart rate measurements using R, G, and B color channels in the facial color analysis have also been related [56,57]. 

## 3. Materials and Methods

### 3.1. Data Collection

Nine university students participated in the study (mean age = 26.88 ± 6.01 years, mean weight = 72.56 ± 14.27 kg, mean height = 172.88 ± 12.04 cm, six males and three females, all white Caucasian). In Table 1 detailed information about the participants is presented. All participants signed an informed consent form prior to data collection and went through protocol familiarization before recordings. The participants were not allowed to talk during the test but could express their feelings freely through facial expression. During video recording, the participants were instructed to look straight at the camera lens. The test consisted of a submaximal ramp exercise protocol in a Wattbike Cycloergometer (Wattbike Ltd., Nottingham, UK), after a 5-min warm up. The initial power output was 75 W, which was incremented in, 15 W min^−1^ until the participants reach 85% of their maximal heart rate (calculated as 208 − (0.7 × age) [58] or be unable to maintain cadence to generate the required power output throughout the stage. Heart rate data were collected at 1 Hz using the Polar T31 cardiofrequencimeter, (Polar Electro, Kempele, Finland) synchronized to the Wattbike load cell for power output measures, sampled at 100 Hz. For the facial tracking, images were recorded during the test using a video camera placed on a tripod approximately one meter from the Cycloergometer in the frontal plane view to capture the participants’ face while performing the exercise, as shown in Figure 1. The camera was adjusted to maintain the angle of 90° between face and camera. The surrounding light source was the fluorescent light bulbs inside the laboratory. The video was recorded at 25 Hz with spatial resolution of 1080 × 1920 pixels. 

### 3.2. Data Procesing

The recorded video was processed off-line to track the average color intensity of a small patch in the participant’s forehead during the exercise. The block diagram of the proposed system is shown in Figure 2. The frame rate was 25 Hz re-sampled to 1 Hz the heart rate was recorded at 1 Hz.

#### 3.2.1. Pre-Processing

Due to the head movement during cycling, some frames extracted from the video recording presented blurring. To overcome this problem Wiener–Hunt deconvolution algorithm was applied in each frame [59], based on a threshold of images variance. After de-blurring, a Gaussian filter was applied for noise reduction.

#### 3.2.2. Face Detection

Participants’ face was detected by Viola and Jones [60] face detection algorithm. The detected face was cropped and resized to 400 × 400 pixels (Figure 3a). Based on the results reported on the literature, forehead patch provides the highest accuracy of heart rate measurement among other patch locations, even better than whole face [41,61]. Therefore, an image of 16 × 16 pixels cropped from whole face in the lower part of forehead patch. (Figure 3b).

#### 3.2.3. Normalization

The images were recorded in a room with lighting background from the AC florescent bulb, where the frequency of supplying electricity was 50 Hz, therefore, affecting the image brightness. The solution to this issue was found in the use of color normalization to eliminate brightness. Consider an RGB image of size *M* × *N* pixels where an image is represented by *I* [*M*, *N*, *c*] where *M* = width, *N* = height and *c* is the color component. The normalized value of each color component for each pixel is calculated by expression 1.
(1)$$I_{norm}\;\left[i,j,c\right]=\frac{I\left[i,j,c\right]}{\sum_{c=1}^{3}\left(I\left[i,j,c\right]\right)},\;\;i=\left[1\;M\right],\;and\;j=\left[1\;N\right]$$
where *c* = {1, 2, 3} corresponding to red (R), green (G), and blue (B) component of image I.

From the above expression, it is common that:(2)$$\sum_{c=1}^{3}I_{norm}[i,\;j,\;c]=1$$

#### 3.2.4. Color Space Conversion

The RGB color model is an additive color system. The combination of the three colors results in visible colors to human eye. Although the RGB color model is the most common for image processing, we added other four different models for tracking color components to compare its efficiency: Hue-Saturation-Value (HSV), Luminance-Chroma: Blue-Chroma: Red (YCbCr), Lightness-Blue: Red, Blue: Yellow (Lab), and YUV. After the normalization of the original RGB image, it was converted to each of the color models.

#### 3.2.5. Patch Averaging

The pixels within the 16 × 16 forehead patch are spatially averaged to yield an individual component of each color space for a frame per second throughout the video frames captured from the beginning to the end of a fatiguing exercise protocol with incremental intensity performed in a cycloergometer and form the raw signals of each channel. After the color space conversion of the normalized RGB, HSV, YCbCr, Lab, and YUV, the patch averaging ${\bar{y}}_{c}\left(i\right)$ was calculated for each color component and space as: (3)$${\bar{y}}_{c}\left(i\right)=\frac{1}{M\times N}\sum_{m=1}^{M}\sum_{n=1}^{N}y_{c}\left(m,n\right)$$
where *i* is color channel such that R or G or B for RGB, H or S or V for HSV and so on. *M* and *N* are the patch dimensions (16 *×* 16) and *y_c_*(*m*, *n*) represents the pixel value of the color channel *i*.

Applying expression 3 for each frame, results in a vector of each component with the length equal to the number of seconds in the video; therefore, for each color model, there are three vectors representing each color component. 

#### 3.2.6. Median Filter

The patch averaging result is usually noisy, thus, to smooth the data, a moving average filter and median filter were applied. The window size was set to 201 points and to solve the endpoint problem, signal padding with the size of 40 points as well as signal reflection were applied.

#### 3.2.7. Average Filter

To smooth the small peaks resultant from the median filter a moving average filter with a small WindowSize was applied. The average value within a sliding window was calculated as expression 4.
(4)$$y\left(n\right)=\frac{1}{WS}\left(x\left(n\right)+x\left(n-1\right)+\ldots +x\left(n-\left(WS-1\right)\right)\right)$$
where *y*(*n*) is an average value, *x*(*n*) is a given value and *WS* is the window size considered for smoothing operation.

The result of the median and the moving average filter is shown in Figure 4.

### 3.3. Statistical Analysis

For all data analysis, both the heart rate and video recordings were re-sampled to 1 Hz. The individual color component of each color space was analyzed separately, such as RGB = [R, G, B], HSV = [H, S, V], YCbCr = [Y, Cb, Cr], Lab = [L, a, b1] and YUV = [Y1, U, V1] as well as a combined form. The normalized heart rate and normalized color intensity of RGB images of one random subject is plotted to illustrate the variation of color intensity with the heart rate changes. Various statistical approaches were used to analyze the color changes with respect to the exercise intensity where exercise intensity was represented by the HR. Most of the statistical approaches are related to the association of heart rate with the color intensities.

The association between HR with the various color models were calculated using continuous analysis of the color intensity. The multivariate time series analysis using multiple variable autoregression with lag of 1 was used to estimate the maximum HR percentage (maxHR%) from the individual color intensities [R, G, and B for RGB model] and its previous values. Before analyzing the multivariate autoregressive model, all the individual independent (three independent variables for each of the participants and each color model) were tested and all the independent variables were stationary. For the time series stationary test, Augmented Dickey-Fuller (ADF) Test were used. The autoregressive model is calculated using the Equation (5). The maxHR% was calculated as HR × 100/(220 − Age). Before fitting regression models, the independent variables were normalized between 0 and 1. The regression models were developed for both the individual participant (personalized model) and the whole sample at a time (universal model).
(5)$$HR\left(t\right)=a_{1}+w_{1}\times HR\left(t-1\right)+w_{n}\times CR_{n}\left(t-1\right)+e\left(t-1\right)$$
where $a_{1}$ is the constant terms, $w_{1}$, $w_{2}$, $w_{3}$ are the coefficients and $CR_{n}$ represents the color values, *e* is the error terms.

A universal multiple polynomial regression model using a Support Vector Regression (SVR) was also used to predict the maxHR% from three components of each color model for all the participants at a time. The root mean square error (RMSE) was calculated for whole sample at a time to find the best predictor of the heart rate through the color components of each of the models. All statistics were performed using dedicated codes written in Python programming V3.5.

## 4. Results

The main objective of this study was to analyze the relationship of heart rate and facial color changes during fatiguing exercise. The results of images pre- and post-processing are shown in respective sections.

### 4.1. Color Intensity vs. HR Plot

The normalized heart rate and normalized color intensity of RGB images of one random participant is plotted in Figure 5. The result indicates that facial color intensity changes according to the effort performed during fatiguing exercise, and this variation correspond to changes in heart rate. Increase in heart rate is related to the increase in blood pump to face vessels, thus, turning face skin redder and combined color intensity get affected. Therefore, increase in heart rate indicates heart pumps to blood vessels more often. However, we observe a decrease in the red component as the heart rate increases. This result can be attributed to the color normalization performed during images pre-processing.

### 4.2. Multivariate Autoregression Analysis 

The multivariate autoregression model was designed and analyzed using combination of the color components at a time. The model was developed to estimate the maxHR% from facial color intensity. The normalized values of all the three components were considered as independent variables for the multiple autoregression. The RMSE of each model for personalized is presented in Table 2. The best average RMSE using individual color component of HSV color model in multivariate autoregression model is 0.255.

### 4.3. Polynomial Support Vector Regression 

Using the same independent and dependent variables used in the autoregression models, a polynomial SVR (degree three) is used to predict maxHR%. The data analysis was limited to the global model instead of the individual participant. The results were interpreted using RMSE, F-values, and R-square values as illustrated in the individual regression model (see Table 3). The significant variation between the individual model and the global model was detected specially in terms of the RMSE. Compared to the autoregression model, the universal polynomial regression model provides inferior with least value of the RMSE. As all the color models were calculated from the RGB color model, the color channel might have multicollinearity between each other suggesting that the independent variable exhibited considerable multicollinearity. The multicollinearity in between each color component of each color model were tested using multicollinearity test and Variance Inflation Factor (VIF) were calculated. The results of multicollinearity test were illustrated in Table 4. All the values of VIF are less than 10 and most of them are less than 5 which indicate that the independent variables are not colinear to each other which suggests to worth enough to develop the global regression model of each color model. 

## 5. Discussion

It is very common that during physical exercise, human body needs more oxygen, that is why, the heart beats faster so that more blood gets out to the body. The heart rate increases in proportion to the intensity of your exercise [62,63,64]. Based on the outcome of the related research studies, it is well known that the exercise intensity and the HR establish the linear relationship. The HR is generally considered as a better parameter for assessing and monitoring relative exercise intensity [5,63]. This study aimed to quantify the relationship between heart rate measured using a contact sensor and heart rate estimation through facial skin color modelling in healthy young people during the sub-maximal ramp exercise protocol. The main findings suggested that the accuracy of estimated the heart rate was higher when facial skin color was modeled (multiple linear regression using fifteen independent variables) using combined color space channels (RGB, HSV, YCbCr, Lab, and YUV), compared to the model (single variable linear regression) that accounted for the individual color channel (R, G, B, H, S, V, etc.).

It has been a while since the change in the use of alternative color spaces in image processing, including in skin color and facial expression analysis [27,28,44,53]. Thus, it was expected that the models would show better performance for face skin segmentation using HSV and YCbCr rather than RGB, as already reported previously [65,66,67]. The green channel of the addictive color space showed the highest accuracy and consistency of the correlation model for heart rate estimation. This is justified by the fact that green light is better absorbed by hemoglobin than red and blue light [67].

As mentioned, there is a significant change in facial color with respect to increased exercise intensity. There is a strong association between the facial color and induced heart rate, as the skin becomes redder due to increased heart rate of physical exertion [45]. However, to our knowledge, this is the first study to demonstrate the use regression analysis to investigate the relationship between facial color and increased heart rate during an incremental intensity cycling. From the results, we found that the strong association between the facial color changes with respect to the increased heart rate. The initial heart rate, maximum heart rate, duration of exercise is quite dependent on individual participant. As our study focuses on both personalized regression model and the global model, the individual regression equation exhibits quite strong association. It is also worth noting that the facial color changes were highly heterogeneous within the participant. 

The possibility of obtaining the global regression model is also investigated and obtained statistically significant results. The data analysis extended to the polynomial regression considering multi-collinearity between the individual independent variables. As Perrett [55] suggested, the facial color changes have strong correlation between exercise intensity. The changes in facial color with respect to heart rate seems nonlinear fashion as the results in global regression equation, suggesting focusing on non-linear regression approaches for future extension of research.

Previous studies have already reported a strong relationship between heart rate and color space components, but mostly for people in resting state, not continuously during exercise [61]. However, because the measurement of intensity is essential to be assessed during physical performance, the determination of the behavior of the color components to predict heart rate in the timeframe enhances the importance of this study. Another aspect worth of notice is on the individuality observed between heart rate and color intensity values. We believe that the current study reaches an important next step by estimating physical effort using image recognition effectively during exercise, nevertheless, it seems that individual variability is still an issue that might be addressed to achieve higher accuracy in further studies.

There are some limitations to this study: (1) Only the healthy participants between the age of 18-36 years were recruited for participation; therefore, the findings of this study may not generalize for other populations such as the elderly. (2) The recruited participants were all Caucasian; therefore, the findings may not generalize to other non-Caucasian peoples. (3) This study was conducted in a laboratory setting in the stationary cycles; therefore, the findings for other types of exercise protocol may not fit into this result. (4) The video was recorded in fluorescent light; therefore, the results may not generalize for other light sources such as ambient and other types of light sources. (5) The number of participants was limited (only nine: six males and three females), and not equally gender-distributed, therefore, it is suggested for further investigation with a greater number of participants with diverse ethnicity and gender. (6) The heart rate is correlated only with the color intensity change but combining other parameters such as temperature might increase the accuracy of HR prediction. 

## 6. Conclusions and Future Work

The experimental results showed that the facial color changes with the increase in exercise intensity. The main findings suggested that the accuracy of estimated heart rate was higher with HSV color model using autoregressive model with the average RMSE of 0.255. Comparing to global polynomial regression model, the individual model exhibits better results. From the overall results, it can be concluded that the proposed models seem best suited for personalized exercise intensity monitoring instead of developing the universal regression model. A high degree of heterogeneity of facial color changes was observed between participants, reflecting individual differences in facial color changes. Exercise prescription and monitoring can benefit from the use of these methods, for example, to optimize the process of online monitoring, without having the need to use any other instrumentation. 

This work can be extended to relate heart rate and the color intensity with more than five-color models, as well as to the investigation of different skin-colors. From the results, it was reported that this method can be appropriate for individual analysis, therefore, this work can be extended by analyzing the data from the same participant in multiple trials and in various situations.

## Figures and Tables

**Figure 1 jimaging-08-00245-f001:**
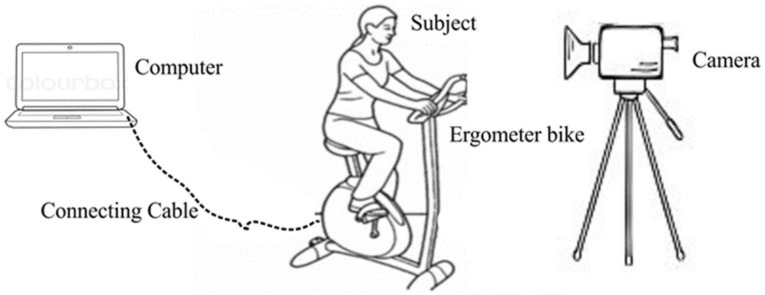
Facial video and HR data collection at the time of physical exercise in stationary cycle-ergometer.

**Figure 2 jimaging-08-00245-f002:**
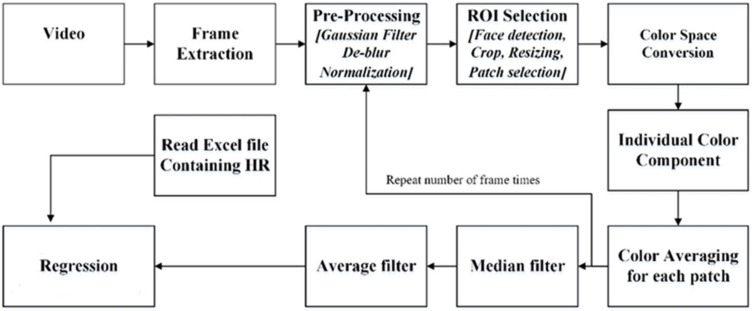
System block diagram of proposed model (ROI-Region of Interest).

**Figure 3 jimaging-08-00245-f003:**
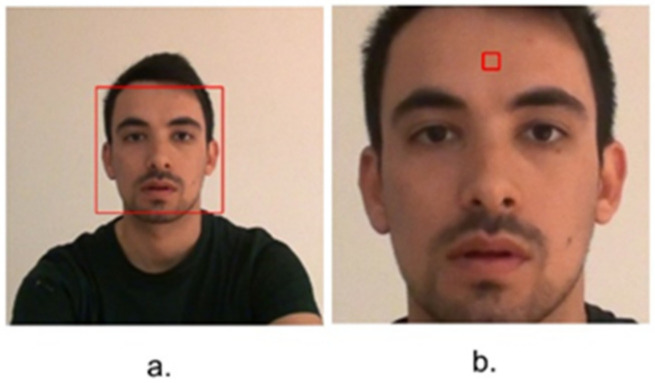
(**a**) Face detection; (**b**) Patch location.

**Figure 4 jimaging-08-00245-f004:**
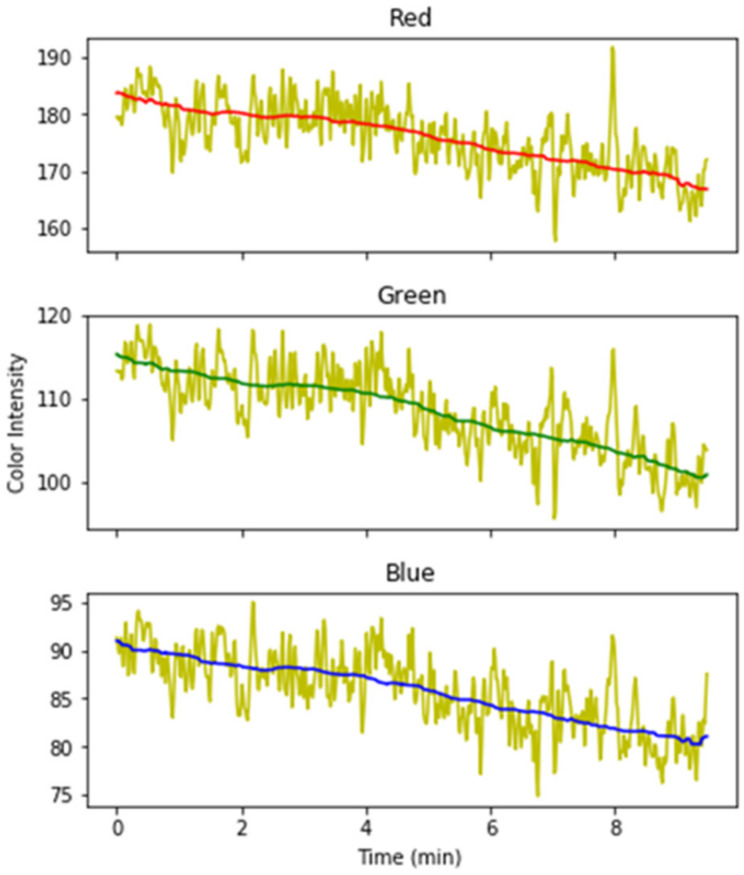
Filtered signals after smoothing operation.

**Figure 5 jimaging-08-00245-f005:**
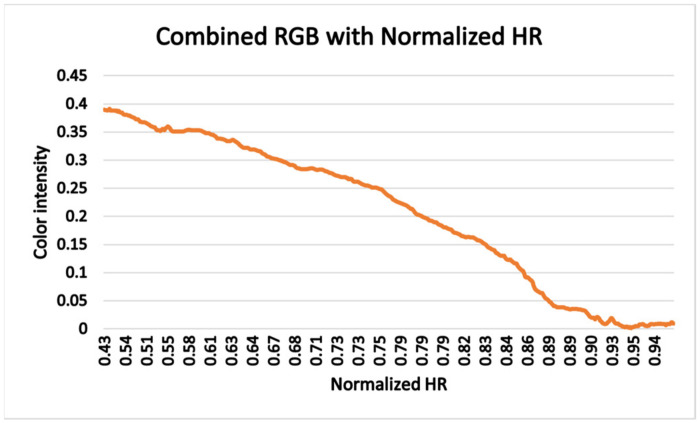
Plot of color intensity value of combined color component of RGB (I = (0.21 × R) + (0.72 × G) + (0.07 × B)/255 × 100 vs. Normalized HR.

**Table 1 jimaging-08-00245-t001:** Physiological information about the participants of the study. The participant’s ID is represented by P1, P2 etc. The initial heart rate was recorded when a participant starts the exercise (after 5 min of warm up exercise) and Final HR (maximum heart of each participant) was recorded at the end of the exercise. HR-Heart Rate, bmp-Beat Per Minute.

Participants ID	Gender	Age (years)	Weight (KG)	Height (cm)	Initial HR (bpm)	Final HR (bpm)	Duration (mm:ss)
Participant 1	Male	22	64.2	172	93	191	9:30
Participant 2	Male	33	66.9	177	70	180	16:00
Participant 3	Female	19	64.2	177	87	191	9:05
Participant 4	Male	36	83.2	182	91	191	15:00
Participant 5	Female	24	66.6	170	97	184	9:20
Participant 6	Female	29	47	157	114	193	8:00
Participant 7	Male	33	83	186	101	180	16:00
Participant 8	Male	22	90.7	195	101	201	12:00
Participant 9	Male	24	87.3	194	115	188	11:00

**Table 2 jimaging-08-00245-t002:** The Root Mean Square Error (RMSE) values using multivariate autoregression with lag 1 between different color models and maxHR%.

Color	Sub1	Sub2	Sub3	Sub4	Sub5	Sub6	Sub7	Sub8	Sub9	AVG
RGB	0.31	0.35	0.42	0.5	0.46	0.35	0.54	0.25	0.24	0.275
HSV	0.31	0.33	0.38	0.35	0.38	0.29	0.51	0.22	0.2	0.255
YCBCR	0.33	0.37	0.46	0.42	0.43	0.36	0.45	0.29	0.31	0.32
LAB	0.32	0.34	0.42	0.45	0.39	0.42	0.59	0.3	0.21	0.265
YUV	0.32	0.38	0.41	0.48	0.48	0.43	0.61	0.23	0.3	0.31

**Table 3 jimaging-08-00245-t003:** Root mean square error, F-values, R-square values for five color models and combination of all the models using a global polynomial regression model with degree three.

Color Model	RMES	F-Value	R-Square Value
RGB	7.85	(F(3,6060) = 4633, *p* = 0.006)	0.70
HSV	6.75	(F(3,6060) = 7360, *p* < 0.001)	0.78
YCBCR	7.84	(F(3,6060) = 3839, *p* < 0.001)	0.92
LAB	7.78	(F(3,6060) = 6905, *p* < 0.001)	0.70
YUV	7.73	(F(3,6060) = 3651, *p* < 0.001)	0.94

**Table 4 jimaging-08-00245-t004:** Multi-collinearity test for all the independent variables of each color model, *p*-values and Variance Inflation Factor (VIF).

Color Model		*p*-Value	VIF
RGB	R	0.0000	2.54
	G	0.0000	9.25
	B	0.00518	8.56
HSV	H	0.2796	1.27
	S	0.0000	1.04
	V	0.0000	1.19
YCBCR	Y	0.0000	3.56
	Cb	0.0000	3.48
	Cr	0.0000	5.14
Lab	A	0.0552	5.24
	a	0.0000	6.14
	B	0.0087	2.85
YUV	Y	0.0000	2.45
	U	0.0041	4.15
	V	0.0000	5.32

## Data Availability

Not applicable.

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
