# Peer review of "Using Computer Vision to Track Facial Color Changes and Predict Heart Rate"

_2313-433X, 2022, doi:10.3390/jimaging8090245_

Round 1

Reviewer 1 Report

Paper deals with important task. The authors tried to develop a non-contact method based on computer vision to determine the heart rate and, ultimately, the exercise intensity

Paper has great practical value.

It has a logical structure. The paper is technically sound. The experimental section is very good.

Suggestions:

1.       The abstract section should be extended using the motivation of this paper

2.       It would be good to add point-by-point the main contributions at the end of the Introduction section

3.       It would be good to add the remainder of this paper

4.       The authors should add a strong Related works section

5.       Is it enoght to use only 9 person and talking about some generalization? Especially if they all was students – yong persons with good skin and without the problems with heart. The authors should explain it

6.       The conclusion section should be extended using: 1) numerical results obtained in the paper; 2) limitations of the proposed approach

7.       It would be good to mention about these papers - DOI:10.1007/978-3-319-63754-9_25, doi: 10.1109/CADSM.2015.7230806.

Author Response

Thanks you for your valuable suggestions. The response to the Reviewer is attached with the revised manuscript along with the revised version of manuscript. Please see the revised sections in the manuscript according to the text color in the response to the reviewer file. 

Reviewer 2 Report

1. Author should explain why HR is the best correlation factor for exercise intensity.

2. Author should explain the intensity of color changes from baseline to target HR in a graded manner. 

3. As forehead is more sensitive area of face to color intensity changes and baseline skin color is variable, individual threshold of changes are also variable. In such scenario, other variable parameter can be combined with color intensity e.g. temperature. Author should provide potential mitigation methods to overcome the limitations of this study.

Author Response

(The authors gave the same response as above.)

Reviewer 3 Report

The idea of predicting heart rate using facial color change is interesting. However, the main issue of this work is the incorrect choice of statistical methods for time dependent data, which makes the conclusions inappropriate to interpret. 

Page 5/Line 171

2.2.1 and 2.2.2 have the same subsection title?

Page 8/Table 2

Both the color components and heart rate data are time series data and are dependent on time. But Pearson correlation is appropriate for independent data. Results from Pearson correlation here could be mis-leading.

Page 9/Table 3 and Table 4

Same problem as the Pearson correlation analysis, data for linear regression models should be independent samples. Results shown here are mis-leading.

Author Response

(The authors gave the same response as above.)

Reviewer 4 Report

I consider that it is an interesting work that contributes to an application of computational vision. Sections 2.2.1 and 2.2.2 have the same name.
It is recommended in table 1 to put what is the maximum heart rate of each of the participants.
How much does light affect, since they are measuring color and this can change due to changes in natural light if it is not controlled.
The work describes the quality of the video and the characteristics such as the adjustment angle for data acquisition.

The preprocessing part is well described, but I consider that a greater description is needed in the data processing.

As future work could be to increase the number of participants and take into account people with other skin types.

Author Response

(The authors gave the same response as above.)

Round 2

Reviewer 3 Report

In general, high VIF indicates that the independent variable is highly collinear with the other variables in the model. Collinearity is a problem for regression models. However, the authors wrote that ‘From the results, all of the independent variables do not exhibit multicollinearity as the Variance Inflation Factor is always greater than 8 and most having values > 100’. The result was interpreted in the wrong direction. If multicollinearity issue is present in the regression model, how would the authors revise the model?

Multivariate autoregressive model requires time series to be stationary, did the author check if this assumption is violated or not?

Rigorous proofread is needed. Just list a couple of them as examples:

1.     Line 247, approchaed -> approaches; analyse -> analyze

2.     Line 252, contineous -> continuous

3.     Line 312, this sentence should be modified if the linear regression analysis has been removed.

4.     Line 321, using al the color color models -> using all the color models.

5.     Incorrect usage of hyphen, e.g. “glob-al”, “de-termination”, “extend-ed”, etc.

6.     This sentence is repeated twice in Discussion: “We believe that the current study reaches an important next step by estimating physical effort using image recognition effectively during exercise, never- theless, it seems that individual variability is still an issue that might be addressed to achieve higher accuracy in further studies.”

Author Response

Dear Sir/Mam,

The response to the reviewer is attached herewith. The corrected part of the article is highlighted with the pink color.

Round 3

Reviewer 3 Report

Line 301: is this 0.11 or 0.255 according to Table 2?

Line 311: inferier with least value of the RMES -> inferior with least value of the RMSE